# OpenReview forum: "Reducing Cognitive Overhead in Tool Use via Multi-Small-Agent Reinforcement Learning"
_ICLR.cc/2026/Conference — Submitted to ICLR 2026_

### Official Review · Reviewer_SjcT · 2025-10-28

**Soundness:** 3
**Presentation:** 2
**Contribution:** 2
**Rating:** 2
**Confidence:** 4

**Summary:**

The paper is argues that single-agent, tool-augmented reasoning systems suffer from "cognitive-load interference" when interleaving high-level reasoning with fine-grained tool operations, which often leads to unstable outputs. To address this, the authors propose a framework that explicitly decouples these responsibilities into a dedicated Reasoning Agent and a specialized Tool Agent. The Reasoning Agent focuses on problem decomposition and planning, while the Tool Agent processes complex tool outputs and acts as an adaptive condenser to manage information flow. To enable collaboration, the agents are trained using a hierarchical reinforcement learning approach with role-specific and collaboration-based rewards, specifically utilizing normalized advantages for the Tool Agent and aggregated advantages for the Reasoning Agent. On mathematical problem-solving tasks involving code execution, MSARL achieves more stable reasoning and higher final-answer accuracy than single-agent baselines.

**Strengths:**

- The paper does a good job motivating the need for multi-agent systems by showing the limitations of single-agent methods in section 3
- The proposed framework demonstrates good performance across math benchmarks with higher accuracy and stability improvements.
- The reward design seems well justified for providing a dense learning signal

**Weaknesses:**

- While the cognitive motivation is strong, the architectural decomposition into a high-level planner and a low-level executor/interpreter is not particularly novel, bearing strong resemblance to hierarchical RL or classic Task and Motion Planning paradigms.
- The study relies exclusively on the Qwen family of models, limiting the generalizability of the findings due to concerns about test-set leakage in proprietary/heavily fine-tuned models.
- The current ablations are too high-level; specifically, the paper is missing a key comparison of the specialized, collaboration-oriented reward mechanism against a standard outcome-only RL approach (e.g., vanilla PPO/GRPO applied globally to both agents).
- While the current experiments use code as a tool, exploring whether this framework extends to a distinct tool-use case like search APIs would provide better evidence that this framework scales and is robust outside of math/code problems.

**Questions:**

- Can the authors provide additional details about the overall distinction between this framing and traditional TAMP frameworks or other perspectives in hierarchical RL?
- Can the authors try their approach with other LLMs to mitigate concerns about data leakage from qwen?
- Can the authors please provide an additional ablation study that directly compares the effectiveness of the collaboration-oriented, normalized advantage reward against a simpler reinforcement learning approach where both agents are trained using only the sparse, final-outcome reward, applying it uniformly to every step in the trajectory?
- How well does this approach work in multi-tool use scenarios?

---

### Official Review · Reviewer_SfFB · 2025-10-29

**Soundness:** 3
**Presentation:** 3
**Contribution:** 3
**Rating:** 6
**Confidence:** 4

**Summary:**

The paper introduces MSARL, a multi-small-agent reinforcement learning framework that separates reasoning and tool use into two collaborating agents to reduce cognitive interference in LLMs. Experiments on mathematical reasoning tasks show that MSARL significantly improves accuracy and stability compared to single-agent RL baselines. Overall, it offers a simple effective architecture for enhancing tool-augmented reasoning.

**Strengths:**

- The dual-agent decomposition is conceptually simple yet effective. The reasoning–tool separation mirrors cognitive modularity and provides a clean interface for information flow.
- MSARL-1.5B achieves 55.9 Pass@1, outperforming both larger (7B) and fine-tuned baselines by up to 5.9 points. Ablations show consistent improvements and stable training dynamics. The inclusion of Pass@8 / Maj@8 metrics demonstrates stability gains.
- The implementation details (datasets, decoding settings, batch, tool-call limits) are thorough and reproducible.

**Weaknesses:**

- All experiments are in mathematical reasoning with code execution. Although the method claims to generalize to “multi-tool” settings, this is not empirically demonstrated.
- Dual-agent training introduces additional overhead. The actual wall-clock or compute cost vs. single-agent RL baselines is missing.

**Questions:**

- How sensitive is MSARL performance to the hyperparameter C (maximum tool calls)?
- Could the Helper agent be frozen while only the Reasoner is RL-trained?
- What is the training cost compared to single-agent GRPO?

---

### Official Review · Reviewer_52CB · 2025-10-31

**Soundness:** 2
**Presentation:** 2
**Contribution:** 1
**Rating:** 4
**Confidence:** 4

**Summary:**

This paper introduces a dual-agent framework that decouples reasoning from tool execution/interpretation for math problem solving with code. A “Reasoner” plans, while a “Helper” parses long tool outputs and returns condensed signals. Training uses SFT on MATH traces, followed by GRPO with role-specific and collaboration rewards.

**Strengths:**

1. The writing is clear.
2. The experimental setup is easy to follow.

**Weaknesses:**

1. The “cognitive overhead” evidence is narrow (three small backbones, N=5 samples, one judge), limiting generality.
2. The method is only demonstrated at 1.5B scale; larger models are not reported.
3. The abstract claims to mitigate hallucinations and to boost tool-invocation tendencies, but there is no quantitative hallucination metric and invocation is capped, preventing analysis of tendency.

**Questions:**

1. Cognitive offload from tool calls may vary with model capacity—do stronger models still require this design?
2. Why choose a multi-agent split instead of a single policy that reads raw tool outputs and emits a compact parse? Please add a single-agent-with-parser baseline on the same backbone.
3. In Table 1, MSARL-1.5B is listed under the “7B-Base” block?
4. The approach appears weak on the challenging AIME benchmark.
5. Does the approach hold across different model scales and architectures?
6. Please include stronger TIR baselines on the same backbone.

---

### Official Review · Reviewer_5ACH · 2025-10-31

**Soundness:** 2
**Presentation:** 3
**Contribution:** 2
**Rating:** 2
**Confidence:** 3

**Summary:**

- Proposes MSARL, a dual‑agent framework that decouples high‑level reasoning (Reasoner) from low‑level tool execution/interpretation (Helper) to reduce cognitive interference in tool‑augmented reasoning

- Introduces a collaboration‑oriented, hierarchical RL scheme (GRPO):
   - Helper gets token‑level advantages from n alternative interpretations per tool use (subgroup rewards).
   - Reasoner gets a trajectory‑level advantage, pooled over those interpretations.
   - Removes KL to a reference model for simpler training

- On math with code execution, MSARL‑1.5B improves Pass@1 vs. single‑agent TIR baselines and shows stronger Pass@8/Maj@8 stability; an “untrained” dual‑agent also helps modestly, suggesting architectural gains beyond learning

**Strengths:**

Originality
- Clear role separation (planning vs. tool interpretation) with a concrete collaboration reward shaping—a principled take on multi‑agent RL for tool use

Quality
- End‑to‑end RL implementation with grouped rollouts, per‑role objectives, and practical engineering (idle‑time mitigation via C)

- Competitive results across AIME24/25, MATH500, Olympiad, AMC23; helpful ablation (trained vs. untrained dual‑agent)

Clarity
- System diagram, roll‑out prompts, and pseudocode (Alg. 1) make the pipeline easy to follow

Significance
- Demonstrates that decoupling tool interpretation can stabilize and improve math/tool reasoning without scaling model size; suggests a scalable blueprint for multi‑tool settings

**Weaknesses:**

Complexity–performance trade-off
- The dual-agent MSARL pipeline introduces substantial architectural and operational complexity (role separation, inter-agent messaging, subgrouped rollouts, tool interpreter integration, global batch size of 1, and a hard cap on tool calls C) without a commensurate accounting of its costs. The paper reports accuracy gains (e.g., +5.9 Pass@1 over the strongest single-agent TIR baseline) but does not quantify end-to-end training/inference latency, GPU utilization/idle time due to agent handoffs, tool-call rates, or cost per correct answer. Without compute/throughput and robustness metrics, it is unclear whether the added engineering, maintenance burden, and potential failure modes (reward gaming via formatting, tool-output brittleness) justify the observed improvements versus simpler single-agent RL baselines with process-level rewards or better tool-use scheduling. Please add wall-clock, GPU-hours, throughput, inference latency, tool-call counts, and multi-seed variance to make the ROI of MSARL’s complexity concrete

Evidence scope and fairness
- Heavily focused on math/code; limited evidence for non‑code tools (retrieval, calculators, web APIs) or non‑verifiable tasks

- Fairness of comparisons unclear: many baselines are single‑agent; report whether they share the same tool budget (C=1), identical prompts, and identical tool runtimes

- Single‑seed reporting; no CIs/variance. Improvements of a few points may be within training variance

**Questions:**

See above.

---

### Meta-Review · Area_Chair_eGC5 · 2026-01-05

**Summary:**

The manuscript proposes MSARL, a dual‑agent framework that decouples high‑level reasoning (Reasoner) from low‑level tool execution/interpretation (Helper) to reduce cognitive interference in tool‑augmented reasoning.

3 out of 4 reviewers gave negative feedback (2, 2, 4), and 1 gave a score of 6.
* Reviewer 5ACH criticizes the lack of quantitative data regarding computational costs, such as inference latency and GPU utilization, to justify the system's high architectural complexity. He/she highlights the narrow focus on mathematical code-use and the absence of multi-seed variance reporting to ensure result robustness.
* Reviewer 52CB argues that the evidence for reduced "cognitive overhead" is too narrow and questions whether the design remains necessary for models larger than 1.5B. He/she also notes the lack of quantitative metrics for hallucination claims and suggests a baseline comparison against a single-agent equipped with a parser.
* Reviewer SfFB points out that the framework's generalization to "multi-tool" scenarios is not empirically demonstrated beyond mathematical reasoning with code. Additionally, he/she requests details on the actual wall-clock training costs and the system's sensitivity to the maximum tool-call limit.
* Reviewer SjcT questions the architectural novelty, noting strong resemblances to traditional hierarchical RL and Task and Motion Planning frameworks. He/she also expresses concerns about potential data leakage from the exclusive use of Qwen models and requests ablations comparing the proposed reward mechanism to standard outcome-only RL.

Given that no rebuttal was provided, the AC recommends rejecting this submission.

**Reviewer Concerns:**

No rebuttals were provided, and the reviewer's concerns were not addressed.

**Reviewer Scores:**

No due to no rebuttal.

---

### Decision · Program_Chairs · 2026-01-26

Reject